# Security Analysis and Improvement of Vehicle Ethernet SOME/IP Protocol

**DOI:** 10.3390/s22186792

**Published:** 2022-09-08

**Authors:** Jinze Du, Rui Tang, Tao Feng

**Affiliations:** School of Computer and Communication, Lanzhou University of Technology, Lanzhou 730050, China

**Keywords:** ICV, SOME/IP protocol, CPN, Dolev–Yao, formal analysis, security evaluation

## Abstract

The combination of in-vehicle networks and smart car devices has gradually developed into Intelligent Connected Vehicles (ICVs). Through the vehicle security protocol, ICVs can quickly realize communication transmission. However, with the more frequent connections between smart in-vehicle devices and the network, the relationship between intelligent cars and external systems is becoming more and more complicated, and in-vehicle networks are gradually facing many security issues. Strengthening the security of in-vehicle protocols has become particularly important. This paper uses the model building method based on the Colored Petri Net (CPN) theory to model the Scalable service-Oriented MiddlewarE over IP (SOME/IP) protocol of the vehicle Ethernet. The security protocol is formally verified and analyzed by combining it with the Dolev–Yao adversary model detection method. After verification, the protocol is subject to three attack vulnerabilities: replay, tampering, and deception. We introduce timestamps and random numbers to strengthen the protocol security. After the final analysis and verification, the improved scheme in this paper can effectively improve the security performance of the protocol.

## 1. Introduction

In recent years, with the combination of in-vehicle device networks and emerging technologies such as intelligent transportation systems and cloud platforms, vehicles are gradually moving towards intelligence and network interconnection [1]. At this stage, the connection between cars has become more convenient and fast, and the relationship between vehicles and other external facilities is becoming closer [2]. While this brings more efficiency and speed to people, the security issues of the Internet of Vehicles are also becoming increasingly prominent [3]. Smart cars’ systems and terminal devices are vulnerable to attacks, and the service applications and platforms that transmit information between vehicles face different security threats [4]. Currently, the protocols applied to the Internet of Vehicles are mainly to meet the reliable communication and transmission function between in-vehicle devices, and there are generally problems of poor security [5]. For example, the traditional Internet of Vehicles protocols: CAN, LIN, FlexRay, etc., have been proven. It is threatened by severe attacks such as network information tampering and virus intrusion [6]. The emerging in-vehicle Ethernet is an organic combination of Ethernet technology and in-vehicle equipment, and its low complexity, high efficiency, and high-cost performance have attracted much attention in the field of car networking [7]. SOME/IP, as a vehicle Ethernet application layer protocol, has also been studied and applied by many scholars. However, like traditional car networking protocols, in-vehicle Ethernet protocols face security issues [8]. These network attacks will directly affect the regular use of in-vehicle equipment and cause user information leakage. In the face of more and more security problems of Internet of Vehicles devices, this paper will select the typical and commonly used vehicle Ethernet protocol SOME/IP as the analysis object. Strengthening the security inside the protocol has essential research significance in the industrial and academic fields.

The main research contents of this paper are divided into the following aspects:1.Use the formal analysis method to analyze the security of the vehicle Ethernet protocol SOME/IP and combine the model establishment and detection method of the colored Petri net theory to design the protocol;2.The formal modeling tool CPN Tools is used to describe the communication process of the security protocol in detail, verify the consistency of its model, add an attacker model to the protocol for intrusion testing, verify and analyze the security, and identify the security vulnerabilities of the protocol;3.A protocol enhancement scheme is proposed to address the protocol’s security vulnerabilities, and the improved scheme’s effectiveness is verified using an attacker model;4.Based on the real vehicle Ethernet environment, the designed security scheme is simulated and simulated, and the performance is analyzed.

The rest of this paper is organized as follows. Section 2 reviews and discusses other scholars’ related work on in-vehicle cybersecurity. Section 3 introduces the preliminary knowledge, which is mainly divided into SOME/IP protocol introduction, CPN Tools, and attack identification and defense. Section 4 models the SOME/IP protocol message flow model and performs conformance verification and evaluation based on the CPN verification tool. Section 5 adds the adversary model and conducts a formal security analysis of the protocol that joins the adversary model. Section 6 proposes a new reinforcement scheme for the security problem of introducing adversary models and the final designs and models of the reinforcement model.F Finally, it compares our model detection method with other works. Section 7 uses the CANoe platform to simulate the protocol and analyze its performance.The last section concludes and presents prospects.

## 2. Related Work

In recent years of research, various scholars have proposed different reinforcement schemes in the face of security analysis of the Internet of Vehicles protocol. Reference [9] proposes a hybrid framework combining blockchain theory to protect the network security in the car. However, this method does not give the specific interaction process of the protocol, nor does it verify the feasibility of its scheme. Reference [10] uses a neural network method to authenticate messages on the vehicle network, such as the traditional CAN protocol. The author claims that it can effectively improve the accuracy of electronic control unit (ECU) transmission and finally gives analysis and verification, but there was no analysis to assess safety. Reference [11] proposes a CAN scheme that can prevent multiple attacks. The author declares that security can be guaranteed without modifying the bottom layer of the CAN protocol. Finally, verification experiments are carried out to show that the proposed method is useful for preventing and defending against various attacks. Reference [12] proposes a security architecture based on the CAN with Flexible Data rate(CAN-FD) network to build a secure transmission environment and a hierarchical encryption transmission technology on the vehicle network that has been verified but lacks a formal verification method to verify its feasibility. Reference [13] analyzed the protocol’s performance through the method of electrical fast transients (EFT) injection and solved the problem of the rise and fall of each EFT injection time in the experiment. However, this method has limitations, focusing only on the in-vehicle communication protocol in a certain type of fault testing, and lacks detection in other related real-time domains. Reference [14] proposed a new authentication scheme to protect the security of the vehicle CAN protocol. Finally, the author verified the proposed scheme to prove its security. Reference [15] proposes a new framework for protecting SOME/IP. Although the scheme gives the theory of security protocol modeling and verification and provides a high degree of assurance of the expected security properties, this method will cause a lot of burden on the network and no intuitive and clear formal verification proof scheme. Reference [16] designs a security architecture for secure transmission between CAN-FD and SOME/IP, and the author states that the scheme can effectively guarantee the integrity and security of data.

In summary, there is a lack of a formal analysis method for evaluating the security performance of in-vehicle networks. There is no specific, formal analysis scheme for analyzing the security improvements in the SOME/IP protocol itself. Thus, it is necessary to analyze the security of its message transmission process using formal modeling analysis.

This paper formally models the SOME/IP protocol communication process based on CPN theory. It uses the CPN Tools modeling tool to extract the key elements to test the protocol’s security. After the analysis and verification visualized by the tool, the protocol itself is found to have security problems. This paper will subsequently propose an improvement scheme to improve the known security vulnerabilities of the protocol. In order to verify the effectiveness of the proposed scheme, security validation of the new scheme is performed.

Compared to recent research schemes, the scheme proposed in this paper has the property of formal verification. This new formal model checking method analyzes the protocol security while verifying the consistency of the original model. In order to study the security vulnerabilities of the protocols, this paper introduces the attacker model. Targeted improvement methods are proposed for the security vulnerabilities found, and the new scheme is verified for security.

## 3. Preliminary Knowledge

### 3.1. SOME/IP Protocol

In-vehicle Ethernet brings more convenience to emerging in-vehicle electronic devices, while the speed and efficiency are greatly improved, the manufacturing cost is also greatly reduced. Many people favor SOME/IP as a high-efficiency protocol on the vehicle Ethernet application layer. Because SOME/IP is a service-oriented and scalable protocol, it is a solution for automotive service-oriented architecture (SOA) middleware for control messages. SOME/IP was proposed by BMW in 2011 and incorporated into the 2014 AUTOSAR specification [17,18]. SOME/IP allows applications to communicate with each other without needing to know which specific electronic control unit (ECU) the application is running on. SOME/IP features include [19,20]:1.Service discovery (SOME/IP-SD): The device uses a dynamic lookup method to find the services the user wants and determines that these services can be used and accessed based on their IP address and device port number;2.Publish/subscribe mechanism: Users can subscribe to the service through the client and quickly obtain the service they want when publishing data;3.Remote procedure calls (RPCs): Use remote method calls to the data and pass data through the network return values.

The data frame structure of SOME/IP protocol is mainly composed of the message header (Header) and message body (Payload). The Header is mainly composed of the following fields:The Message ID is used to uniquely identify the message, including Service ID and Method ID, each of which occupies 16 Bit;The Length symbol represents the length of the message, which starts from the Request ID of the Header and ends at the end of the Payload;Different Request IDs are called to distinguish the same message, mainly composed of Client and Session IDs;The remaining Protocol Version represents the version number of the protocol header. The Interface Version represents the interface’s version number, the Message Type is used to indicate the message type, and the Return Code is used to determine whether the request message is successful.

The message body Payload is the payload, which contains the content of the message. Usually, its length is not fixed.The specific protocol data frame structure described above is shown in Figure 1.

In order to ensure the interoperability of the SOME/IP protocol, the format of all protocol headers is consistent. The fields in the header are displayed according to the transmission order. The fields in the upper left corner are first sent, and the order to be processed is also determined by the position of the fields in the message. Therefore, unicast, multicast, and broadcast are supported, and different communication types can be applied to different communication scenarios. At the same time, it is scalable and suitable for small and large platforms, such as automobile subsystems and whole vehicle systems.

### 3.2. Message-Oriented Middleware

As a middleware protocol, SOME/IP has some characteristics of middleware. After the service provider of the middleware implements the function, the service-related information will be uploaded to the service registry [21] through the predefined standard service description format. The service requester queries the registry when needed, obtains the relevant service interfaces and usage methods, completes the remote invocation operation, which is not limited by the development platform, programming language, and software architecture, and maximizes the independence between service consumers and service providers. Figure 2 shows the working principle of middleware.

In conclusion, as more and more automotive devices adopt SOME/IP to facilitate communication between processes and devices, security analysis of the SOME/IP protocol becomes critical.

### 3.3. CPN Tools

CPN Tools [22] is an open tool that integrates editing, design, and simulation advantages. It can dynamically and intuitively present the interaction process between each module to the user. It has rigorous mathematical logic and can refine every step of the running process of the established model [23]. Through the tools that come with CPN Tools, such as creating model tools, simulation tools, hierarchical layering tools, and analyzing state space tools, we can formally analyze the correctness and rationality of the model. Using internal auxiliary tools can simulate and debug the message transmission process of the protocol in any field, and the markup language (ML) used by it is also easier to understand. The boundedness and reachability analysis of the final state space analysis can also reflect the model problem in detail.

### 3.4. Attack Identification and Protection Measures

This section will introduce several man-in-the-middle attacks in detail, how to identify whether they are attacked, and the corresponding defense measures.

#### 3.4.1. Attack Identification

As an illegal third-party participant, a man-in-the-middle attack significantly threatens the normal interaction process of both communication parties. Severe cases can threaten the entire security system and cause server outages, thereby committing criminal offenses. Attackers often target financial gain, using standard techniques such as session hijacking and Domain Name System (DNS) spoofing attacks to compromise websites [24].

There are methods and tools available for man-in-the-middle attacks to identify and detect security threats. For example, when an attacker conducts an attack, an additional time delay will be generated in the average communication time. At this time, it can be judged whether it is attacked by checking its sending response time difference. Secondly, we can analyze whether there are abnormal data packets. If we find a big difference from routine data, we can suspect that we have been attacked. For many Internet of Things (IoT) devices, using intrusion detection is also a good choice. Capturing abnormal traffic data and analyzing the difference from standard data can be used to identify initial attacks.

Many scholars have also given different solutions for attack identification: Reference [25] uses the method of analyzing network traffic data to identify man-in-the-middle attacks in industrial systems. Reference [26] uses fog computing to identify security vulnerabilities in IoT devices. In our scheme, we judge the security problems of the protocol through the scheme of model meta language (ML) language recognition and detection.

#### 3.4.2. Defense Method

There are many ways to defend against an attack. For example:Improving inappropriate two-way authentication to prevent attacks can prevent attackers from stealing internal information;Secondly, a more internal solid management system can be established; once an attack is found, managers can quickly find and deal with it effectively;Enhancing the complexity of the verification required for authentication is also a good defense;Authentication with an authentication certificate issued by a professional authentication agency prevents eavesdropping and sniffing by attackers;A particular secure communication channel is established to verify the data exchanged between the two parties, and if a data leak occurs, it can be quickly detected;Finally, the user’s safety awareness should be strengthened, and illegal operations should be paid attention to daily use.

## 4. SOME/IP Protocol Modeling

### 4.1. Protocol Message Flow Model

Due to the complexity of the SOME/IP protocol system itself, it is too cumbersome to use the traditional CPN modeling methods in the past. Therefore, we modularize the entire complete model into small parts during modeling. Using this idea, the entire modeling process will be divided into different levels to complete it entirely. To build a hierarchical model, first use the alternative transition tool of CPN modeling to build the top-level network model, split the top-level network model into different sub-pages—these different pages represent different interactions between the underlying network models—and then use the alternative transition’s association effect. The process of gradually improving and refining the underlying models of each alternative transition is the process of gradually completing the entire system modeling. The descriptions of the symbols expressed in this paper are shown in Table 1.

The message flow process of SOME/IP is divided into several steps. The specific steps are as follows:The message flow model of the protocol is mainly divided into authentication, the key negotiation phase, and the message sending phase. The operations to be performed by Controller A, Controller B, and the Ethernet gateway during these interactions are described in detail in Figure 3.Suppose DCA wants to establish communication with DCB. The DCA first initiates a connection request and encrypts the ID of the DCB with the key KA generated by itself, obtains the encrypted ciphertext (IDB, MIC) KA, and sends a communication request to DCB.After receiver B receives the communication request from sender A, it judges whether the ID is legal and checks whether the message is complete. Otherwise, the DCB will require retransmission, and the request message will be decrypted if it is legal. The next step is to send an authentication request to the gateway SSC. The request message includes the IDA of the DCA, its ID, and the integrity verification code MIC.After receiving the verification request from the DCB, the SSC determines whether the identifiers of IDA and IDB are legal. If the request message is illegal, the SSC will discard the illegal message. Otherwise, it will continue to the next step. SSC uses the two keys—KA and KB—to decode the ciphertext to obtain IDA and IDB. Then, the consistency of the IDA and IDB obtained by decryption and the IDB obtained by plaintext is verified. If inconsistency and authentication fail, the message is discarded; otherwise, the process continues to the next step, which is too generate a temporary session key SK between DCA and DCB. The IDB and the connection message MES are encrypted using the key KA. IDA and MES are encrypted with key KB, and then a response message is returned to DCB. The message contains ciphertext (IDB, MES) KA, ciphertext (IDA, MES) KB, and session key SK.After DCB receives the response message from SSC, it decrypts the ciphertext with the key KB and verifies the consistency between the decrypted IDA and the IDA in the communication request message. The SK will be saved as a temporary session key if the verification is successful. SK is used to encrypt IDA and IDB to obtain the ciphertext (IDA, IDB) SK. Finally, a response message is returned to DCA.After receiving the response message from DCB, DCA decrypts the ciphertext with the key KA to obtain the IDB. The consistency of the IDB obtained by decryption and the IDB being communicated is verified. After the verification is successful, DCA continues to the next step, which includes decrypting (IDA, IDB) SK, obtaining IDA and IDB, and verifying. The SK will be saved as a temporary session key if the verification is successful.After DCA and DCB obtain the temporary session key, secure transmission is performed. DCA will use the session key SK obtained when the session key is established to encrypt the transmission data DATA and its own IDA, and the transmission message includes (DATA, IDA) SK.After DCB receives the sent data, it verifies whether the ID is legal and whether the message length is consistent. If it is legal and consistent, the counter will be incremented by 1. Otherwise, it will be discarded and retransmitted.

### 4.2. Modeling of the SOME/IP Protocol CPN Model

In the model of establishing this protocol, it mainly includes three parts: device controller A, device controller B, and Ethernet gateway. In order to reduce the complexity of the model, while simplifying the model, we must also consider that the accuracy of the protocol communication process cannot be reduced. In this paper, the hierarchical modeling method will be adopted, and the model will be divided into two layers: the top layer and the bottom layer. Finally, after classifying each layer, the sub-transition corresponding to the underlying substitution transition is described in detail.

When modeling the message flow mechanism of the SOME/IP protocol, we will use the model creation tools that come with the CPN tool, such as transition and place, to model the entire process [27], and the specific interaction process is represented in the form of a model. The top-level model will be used as the overall conversion process of the protocol, simulating the communication initiator, communication receiver, and Ethernet network of the protocol, and the double rectangle represents the replacement transition. The ellipse represents the message repository, the initiating device of communication is represented by DCA, the receiving device of communication is represented by DCB, and the secure Ethernet network manager of communication is represented by SSC. The top-level model of the protocol is shown in Figure 4, which includes three alternative transitions and five message places. The request connection and connection response is initiated by the communication device initiator DCA to the gateway SSC, and the process of obtaining the session key is represented by S1, S2, S3, and S4. After the connection is established successfully, the sender obtains the session key and performs security transmission. The communication sender sends the secure data to the receiver, and S5 represents the secure transmission data.

The underlying model of this protocol consists of three parts. Firstly, according to the message flow mechanism of the protocol, the whole protocol is in order of session request and response and then establishes the data transmission connection process. Finally, according to the way of transmission, the process of its secure transmission through two controllers with the same session key relationship is explained in complete detail.

The request connection process and the connection success process occur between the three alternative transitions—DCA, DCB, and SSC—and these three underlying subvariants can be simulated in complete detail for their transmission paths and response message paths, respectively. The data security transmission path occurs between the alternative variants DCA and DCB, and we can also describe their processes in detail to simulate the model.

Figure 5 details the internal structure model of the alternative transition DCA. The transition MES integrates the key KA internally generated by the sender DCA and the receiver’s ID. The transition MEs integrate the message encrypted by the key KA and its own ID into a request message, and finally, the transition Reqmes integrates it into an agnostic request connection and sends the message to the receiver through the repository S1. After the identities of the communicating parties are legal, KA and KB keys are used to encrypt the session key SK and ID identification, and then they are sent to the library port S3 in the form of a response message. After receiving the message, the receiver DCB uses the key KB to encrypt the message. Decryption is performed to obtain the session key SK and save the session key. After both parties of the communication obtain the session key, the next step is to perform secure transmission, and the transition sendDATA integrates the sent data content into the content of the secure transmission message, including header bytes. Moreover, the data packet, transition safemes, re-integrates the data and counts it as encrypted by the session key SK and sends it out through the S5 port of the place.

Figure 6 shows the internal model of the alternative transition DCB. After receiving the request message sent from the port repository S1, the receiver DCB first verifies whether the ID of the sending requester is legal. If legal, keep the request message sent by the sender. The communication device DCB internally generates a key KB and uses the generated key to encrypt the ID identification. The retained messages are retained with self-encrypted request messages by transitioning CONmes. Finally, the integrated request message is sent to the secure Ethernet gateway device through the output port repository S2; the DCA receives the response message sent by the DCB through the transition REcMES and uses the key KA to decrypt the message content, including the requested session key SK. The session key SK is saved and used to decrypt the message to obtain the content of the message. The repository ReqMES2 integrates two response messages that the Ethernet gateway will send to the two communication devices, including the session key SK and ID identification required for secure transmission. Finally, it is sent to the communicating party DCB in the form of msg3 through the place port S3. The place port S5 receives the message msg5 from the secure transmission party, and the receiver uses the session key SK to decrypt the message. After obtaining the content, verification of whether the ID identification is legal and whether the length of the message conforms to the standard is carried out. If all are as expected, the counter will be incremented by one. Otherwise, the packet will be discarded.

Figure 7 shows the internal model of the alternative transition SSC. The Ethernet gateway receives the request message sent from the DCB through the place port S2, and the transition RecMES transfers the decomposed message and uses the key to decrypt the message to obtain the message content. The transition IDsend connects the Ethernet gateway and the communication. The response message sent by the device is integrated, and the response message is sent to the controller DCA in the form of message msg4 through the output terminal S4 of the place.

### 4.3. SOME/IP Protocol Model Conformance Verification

The accuracy of the original model reflects whether the function of the protocol conforms to the specification and determines the evaluation and validation of subsequent models. In this section, we will use the state space analysis tool that comes with CPN Tools for verification. The purpose of verification is to verify whether the model we established has obtained the expected results through theoretical analysis.

#### 4.3.1. Analyze Expected Results

For the established CPN model, we analyze its state, mainly to study the active state of state nodes and transitions, and then we compare it with the expected protocol state to judge whether it is accurate. According to the communication process of the protocol in Figure 3, after the two parties of the communication complete the identity authentication, they will conduct secure communication later. At this point, the receiver will trigger all transitions, so there will be no dead transitions. Secondly, from the beginning to the end of the transmission between the two communication parties, it will be reflected in the running state of the model, and there will be no running stagnation during the period. Therefore, it can be expected that there will be no live transitions and only one dead state node.

#### 4.3.2. State Space Results Analysis

In this section, we mainly use state space analysis tools to analyze the property capabilities of the protocols in the model. The primary purposes of the analysis are:Verify that there is no deadlock condition in the model;Determine the final state of the model;Determine if the transition has a reachable state to enable;See if there are transitions that are always executing;Find if there is a reachable path from one marker to another.

Table 2 gives the state space query results of the original protocol model.

By analyzing the results in Table 2, it can be concluded that in the entire message interaction process of our model, each transition node is a reachable state. The request messages that exist on the node are all runnable, and the interaction endpoints of both parties are unique. It can be seen from the table that the number of state space nodes, directed arcs, strongly connected nodes, and the number of strongly connected arcs are the same, which indicates that our model does not have an infinite state loop and iterative behavior. The number of dead nodes is one, indicating that there is no request that cannot be executed. In the process of message node transmission of the protocol, the endpoint of the node is uniquely determined at any time. Dead and live transitions are zero because there are no unreachable transitions and transitions are always in cyclic transmission in the model. Through the above analysis, it can be verified that the protocol modeling is consistent.

## 5. Add Attack Model Evaluation and Verification

### 5.1. Dolev–Yao Attacker Model

Dolev and Yao published an article on the importance of discussing the security properties of the protocol [28]. This article mainly proposes that based on considering the absence of flaws in the cryptographic mechanism of the protocol, it first analyzes whether the internal logic of the protocol is reasonable and then discusses its own security properties. The advantage of this method is that it avoids the complexity analysis of the algorithm and can directly explore the security performance of the protocol itself. The Dolev–Yao attacker model proposed on the basis of this theory has powerful attack capabilities. The main attack capabilities are as follows:**Illegal tampering:** Attackers use illegal means to delete or modify legitimate data;**Eavesdropping:** The attacker listens to the messages transmitted by the user without being noticed;**Illegal interception:** The attacker illegally intercepts the transmission message and saves it without being discovered by the protocol;**Replay:** The attacker modifies the previously illegally intercepted message and sends it repeatedly to both parties.

The Dolev–Yao attacker model conforms to the hierarchical modeling idea of CPN Tools. It can act as a protocol participant and affect the entire communication process of the protocol as an attacker at any time. These characteristics are in line with the research purpose of this paper. In the next section, we will use the attack capability of the attacker model to conduct a modeling analysis.

### 5.2. Modeling Based on Dolev–Yao Attacker Model

This section uses the Dolev–Yao model to evaluate and verify the security of the protocol itself, combined with the powerful attack capabilities endowed by the Dolev–Yao attacker model [29], including malicious attacks such as eavesdropping, replay, and tampering, as described in the previous section. At the same time, the model is used to encrypt, decrypt, split, and merge the information sent during the protocol transmission process [30].

We assume that the established model does not consider the security issues of cryptographic algorithms and that it only discusses the security characteristics of the analysis protocol itself. The SOME/IP protocol is a real-time protocol. The transmission time between communication devices is very short, and the encryption, decryption, and verification processes are implemented by the device itself for each data exchange. The receiver joins the attacker.

According to the capabilities of the Dolev–Yao adversary model, a man-in-the-middle attack is introduced on the transmission channel of our original model. Figure 8 shows the introduction of man-in-the-middle attacks, including replay, spoofing, and tampering attacks, to the original model. As shown in the figure, the place and transition marked in red simulate a replay attack, and transition B1 intercepts the first request message of the protocol. Places A2 and A3 store the decomposed message and the message to be decomposed obtained at transition B1, respectively. Transition B3 uses the attacker’s decomposition ability to decompose and save the message in place A6. Transition B4 uses the attacker’s split-combination function to store messages in place AB. Transition B5 saves undecipherable information to Merge. Place A7 initiates the compositing state to go through the merging and decomposing cycle again only after each decomposing message is generated. Transitions B8 and B9 synthesize and send the attacker’s message to the output port place S2. The message arcs, places, and transitions marked in purple in the figure represent the tampering attack of the simulated attacker. The message output arc tamper sends the tampered message to the port place S3. The blue part in the figure simulates spoofing attacks, including transition RE-MES and transition Reqmes2 in the original model.

### 5.3. Security Evaluation of SOME/IP Protocol

In Section 5.2, we add the attacker model shown in Figure 8 to the original model. This section will evaluate and verify the security of the protocol message transmission process, use the state space tool of CPN to obtain state data, and analyze the change process of the data to summarize the existing security problems.

#### 5.3.1. Attacker Model Consistency Analysis

This section performs functional consistency security verification on the model we built in order to verify whether the model’s consistency meets the requirements. Whether the consistency is satisfactory or not is to judge whether our model is appropriate or not, and it becomes particularly important to analyze the state data. Table 3 shows the state space report after adding the attacker model, as shown in Figure 8. The REY-ATK column represents the state space report generated after using the replay attack model. TAR-ATK is listed as a state-space report generated after using the tampering attack model. The SPF-ATK column is a state-space report generated after using the spoofing attack model. The Attack model is listed as the state space report obtained after using the three attack models.

From the data in the table, it can be seen that the number of state space nodes, state space directed arcs, and strongly connected nodes are the same, which means that all state nodes of the attacker model are reachable, and there is no infinite loop or iterative state behavior. In addition, it can be seen from the table that after introducing the adversary model, the number of state space nodes and state space directed arcs increases significantly, but no state space explosion occurs, which achieves the expected results. After the introduction of the attacker model, the size and number of nodes in the state space reduce the messages desired by both communicating parties. These features significantly improve the efficiency of the attack model, and the attack capability can also be guaranteed, which finally shows that the attack model is efficient.

#### 5.3.2. Security Verification

In Table 3, comparing the REY-ATK replay attacker model with the original model, it is found that the number of state space nodes and directed arcs rapidly increases. Attackers use their attack methods to generate a large number of repeated messages, which results in a significant increase in the state space. At the same time, it can be found that there are eight dead nodes because the receiver node has discarded a large number of repetitive messages. The TAR-ATK tampering attack will illegally tamper with the verification information, which results in the fact that the receiver cannot be legally verified, and the number of dead state nodes has increased compared to the original model. For the SPF-ATK spoofing attack model, there are six dead state nodes in Table 3, indicating that the attack behavior leads to an unknown result in the protocol. The encrypted transmission method of both parties of the communication is to use the session key and carry the authentication key. When the attacker implements the spoofing attack because the shared key cannot be obtained, the protocol will not operate normally, so the above six dead state nodes are generated. Comparing the data of using three attack methods at the same time, our model data reflects that the state changes in these three attacks are correct and consistent. The introduced adversary model effectively attacks the request and response information transmission process of the original model, reflecting the existence of three attack vulnerabilities—tampering, deception, and replay—in the original protocol.

#### 5.3.3. Protocol Vulnerability Analysis

Through the analysis of the state space in the previous section, we found that in the first three attack states, the attacker can send a legitimate message authentication request to the receiver without knowing the session key. The receiver still stores the wrong information and sends the message normally without knowing it. Since the protocol sends messages in clear text, the request and response information may be illegally intercepted and tampered with, which will eventually destroy the verification of standard messages. This leads to the following types of attack threats:If an attacker changes the value of the message sequence number, the protocol will be authenticated incorrectly during authentication, and the protocol may fail to authenticate;By intercepting plaintext information, once the attacker obtains a message with the same sequence number, he can replay the message authentication code intercepted in the session to the receiver, which will affect subsequent normal sending and receiving commands;Since there is a plaintext message when the two parties send the message, the attacker will eavesdrop on the communication data, analyze the illegally obtained data, and launch an attack on the node.

## 6. Protocol Improvement and Hardening

### 6.1. Protocol Enhancement Scheme

Through the above security analysis and evaluation, we found that there are many security vulnerabilities in the introduced attacker model. In response to the analysis results, we reinforced the request and response communication process of the protocol. To ensure the real-time nature of the sent data, we added a timestamp mechanism on each sent message segment and a maximum security threshold for the verification time in the communication party; a random number was added in the authentication phase of both communication parties to ensure that the data were securely verified. Figure 9 shows the message flow of the hardening scheme.

The transmission process after hardening is as follows:The requestor initiates a connection request, and the request message contains the transmitted timestamp information, the ID identification of both sides of the device, and the random number generated by the requester.After receiving the communication request from DCA, DCB determines whether the identity is legal and whether the timestamp meets the requirements. If the request message is invalid, DCB will discard the request message to stop authentication; otherwise, it will proceed to the next step and continue to send the authentication request to the gateway SSC. The request message contains the ID of both parties, DCB KA, IDA KB, timestamp, and a random number.After receiving the authentication request from the DCB, the SSC determines whether the timestamp meets the requirements and verifies whether the identity identifiers of the two communicating parties are legitimate. If the request message is illegal, the SSC will discard the illegal message. Otherwise, it will continue to execute the next step, which is to generate a temporary session key SK for DCA and DCB communication. Then, it will return a response message to DCB, which contains cipher(IDB, SK)KA, cipher(IDA, SK)KB, timestamp message, and a random number.After the DCB receives the response message from the SSC, it determines whether the timestamp meets the requirements. Otherwise, it discards the reset, and if it does, it decrypts the ciphertext with the key KB to verify the consistency of the decrypted ID and random number with the ID and random number in the communication request message. If the verification is successful, the SK will be saved as a temporary session key. A response message is returned to DCA. The message contains the ciphertext (IDB, SK)KA, ciphertext (IDA, IDB)SK, timestamp, and a random number.After receiving the response message from DCB, DCA determines whether the timestamp meets the requirement. If yes, it will decrypt the ciphertext with key KA; otherwise, it will discard the reset. Afterwards, it will verify the consistency of the ID and random number obtained by decryption. After successful verification, DCA continues to the next step: decrypt (IDA, IDB) SK, obtain IDA and IDB, and verify. If the verification is successful, SK will be saved as a temporary session key.The DCA and DCB obtain a temporary session key for secure transmission. The DCA will encrypt the transmission data DATA, as well as its own IDA using the session key SK obtained at the time of session key establishment. The transmission message includes (DATA, IDA) SK, a timestamp, and a random number.After the DCB receives the sent data, it will judge whether the timestamp and random number meet the requirements; if the verification is successful, it will continue to verify whether its ID and message length are legal. If they are legal and consistent, the counter will be added 1. Otherwise, it will be discarded and retransmitted.

### 6.2. Protocol New Solution Model

In response to the discovered protocol security vulnerabilities, the original protocol was hardened with a new scheme and validated with CPN modeling. Figure 10 shows the internal model of the improved new scheme DCA. We add a place representing the timestamp by T to the original model, which increases the operation time and transmission time with each message transmission variation. In order to limit the time of each message transmission, we add a time limit, and once we find that the time limit is exceeded, it means that the message is compromised or lost. As shown in the figure, the variation MEs integrate the sent messages, including the newly added timestamps and random numbers, and send them to the recipient for verification. The receiver accepts the response message through the S3 place port and verifies whether the condition is timeout by judging the timestamp of the changeover RecMES2. If the message is timeout, it will be discarded and retransmitted. The message with the timestamp and the random number is integrated by the transition safemes and sent to the message receiver for validation through the place port S5.

Figure 11 shows the improved SSC internal model. As shown in the figure, the communication receiver receives the message from the sender through port S2 and judges whether the message has timed out through the transition RecMES2. If it times out, it will be discarded and retransmitted. The transition IDsend adds timestamps and random numbers through integration and then encrypts the message with its own private key. The transmitted message includes the identity of both parties and the session key. Transition Resend1 encrypts the message as msg4t and finally sends it to the receiver through the port repository S4.

Figure 12 shows the structure of the modified DCB internal model. As shown in the figure, the receiver receives the message sent from the sender through the port repository S1, and the variant RE-MES determines whether the time is timed out and verifies whether the device ID identification is in compliance with the law. If not, it is discarded and retransmitted. The variant CONmes integrates the encrypted message with the timestamp and random number and sends it to the authentication center gateway for request message verification through the port output library by S2. The receiver receives the response message through S4; the library REcMES verifies whether the timestamp meets the security requirements; and the device side verifies whether the device ID and random number are legal by verifying the message obtained by decryption and then discards the retransmission if it is illegal. Transformation ReqMES2 integrates the message, with the timestamp and random number, and sends it to the receiver via output port S3. The receiver’s receive port library S5 receives the message to verify the timestamp, compares the random number with the identity, and the successful verification message counter plus 1 is returned to the sender.

### 6.3. Improving the Program Security Assessment Model

In this section, the same Dolev–Yao adversary model is used to attack the improved communication hardening model of the new scheme. As shown in Figure 13, the part marked in red simulates the replay attack, the part marked in purple simulates the tampering attack, and the part marked in blue simulates the spoofing attack.

### 6.4. Security Evaluation and Analysis of Improvement Scheme Models

In the previous section, we added three attack models. This section will analyze the changes in the space report after adding these types of attacks in detail. Finally, the security performance of the improved protocol is analyzed.

#### 6.4.1. Security Assessment in Three Attack Environments

Table 4 presents the state space data of the three attack models before and after improvement. Among them, REY-ATK represents the replay attack model, TAR-ATK represents the tampering attack model, and SPF-ATK represents the spoofing attack model. From the data in the table, we can find that the number of state nodes after the improvement is significantly reduced compared with the number of directed arcs before the improvement, and the performance of this change is in line with our expected results. Comparing the changes in the REY-ATK replay attack status data before and after the improvement, it is found that the number of dead nodes is reduced from 8 to 2, which indicates that the replay attack method no longer has a good attack effect under our improved scheme. Compared with the TAR-ATK tampering attack, the number of dead nodes has changed from two to one. When the attacker wants to tamper with the communication data between the two parties, but the session key of the communication between the two parties cannot be obtained, the tampering method cannot continue to be implemented. The state of dead nodes in the improved SPF-ATK spoofing attack is also reduced from the original six to one, which indicates that after adding complete authentication, the attack behavior will not continue to lead to an unexpected state of the protocol.

#### 6.4.2. Improved Program Safety Assessment

As shown in Table 5, by comparing the state space analysis results before the improvement, the number of nodes and arcs in the improved state space is significantly reduced, and the number of existing dead nodes is also reduced from 11 to 2. This shows that our improved hardening scheme is effective, and using three attack methods at the same time cannot cause an effective attack in our improved scheme. This is because timestamps and random numbers are added to our scheme, which can effectively resist replay, tampering, and spoofing attacks. Illegal messages will not be legally permitted by the communication device during the verification process. If there are a large number of illegal messages, the means of attack, through the detection of the device, will only be discarded.

The analysis shows that the attacker cannot launch the attack because the attacker cannot obtain the session key, the authentication key, and the complete message of the data, which proves that the reinforcement scheme can resist these three kinds of attacks and conform to the security mechanism of the protocol.

#### 6.4.3. Security Analysis of Improvement Schemes

This section analyzes the security of our improved scheme based on the attacker model mentioned in Section 5.1 and the state space data in Table 4 and Table 5. Details are as follows:In our improved scheme, each time the communication parties DCA and DCB perform message transmission and identity authentication, they must use the private keys KA and KB for encryption and decryption as well as random number authentication. After the final authentication is completed, the session key of the authentication center is obtained. These measures ensure the legality and security of communications. The attacker cannot monitor the transmitted data in this environment, so the improved scheme has anti-monitoring ability.In our improved scheme, the DCA and DCB of the two parties in the protocol communication will add timestamps when verifying and transmitting messages. At this time, if there are a large number of repeated and wrong messages, they will be judged as illegal messages during verification. This ensures the freshness of the message, making it resistant to replay attacks.During communication, both parties will generate random numbers, which ensures that the session key SK generated each time a message is transmitted is correct and legal. Even if the previous session key is leaked, it will not affect the next communication process. Thus, our scheme has forward security.In our scheme, the key is stored for each encryption and decryption. If the attacker forcibly modifies the ciphertext message when the key cannot be obtained, the receiver cannot identify it, which will cause the normal function of the protocol to fail. It is very difficult for an attacker to use masquerading to obtain legitimate information, so our scheme can prevent masquerading attacks and man-in-the-middle methods.

### 6.5. Improvement Scheme Performance Analysis and Method Comparison

In order to verify the effectiveness of the scheme proposed in this paper, in this section, we compare the proposed protocol reinforcement scheme with other high-security schemes, and the results are shown in Table 6. Our scheme adds a timestamp and random number mechanism, which improves the security problems of the original protocol, lacking real-time performance and being vulnerable to tampering. The analysis shows that in the process of communication transmission, the state space of the reinforcement scheme is significantly reduced, the security performance is also significantly improved, and the calculation time is also greatly reduced. Reference [31] uses Tamarin to analyze the SOME/IP protocol formally. The analysis results show that the protocol has three different types of man-in-the-middle attack vulnerabilities. In order to prevent these attacks, a symmetric cryptographic encryption scenario scheme is proposed, and its feasibility is verified. Reference [32] designs an authentication scheme in the vehicle domain, and the security analysis of the Proverif analysis tool shows that the designed scheme can effectively resist malicious attacks. The model detection method proposed by us can detect the security anomalies of the protocol, and through the model, it can more intuitively show what security vulnerabilities exist in the protocol under what kind of attack type. The model checking method is also applicable to the analysis and verification of other in-vehicle network security protocols.

## 7. Improved Protocol Performance and Analysis

### 7.1. Performance Evaluation Based on CANoe Platform

In the previous section, we investigated the security of the improved protocol and performed analytical evaluations and validations; however, these analyses were not incorporated into a real in-vehicle network. Therefore, this section will simulate and simulate our scheme based on the real vehicle Ethernet environment and the experimental platform, as well as analyze and evaluate the performance of the scheme according to the data results. CAN open environment (CANoe) [36] is a bus development tool launched by the German Vector Company. Its main functions include simulation, test analysis, and development. It can not only simulate and analyze the traditional CAN bus based on the real network environment, but also analyze almost all vehicle network protocols [37], including vehicle Ethernet, so it is widely used by many developers and scholars. We leverage CANoe’s powerful capabilities to perform design analysis based on our solutions and SOME/IP protocol specifications.

Figure 14 shows the protocol simulation environment we designed, which includes two communication nodes and an Ethernet gateway. The Ethernet standard database is added through the database function module Databases of the platform, and the corresponding environment variables and system variables are set to associate the two communication elements. When we make relevant operations, there will be corresponding input and output responses. This can ensure that the environment model we design is in line with the standard requirements and, at the same time, does not lose authenticity and generality. In addition, this model has a certain generality and is also applicable to other in-vehicle Ethernet protocols.

Since SOME/IP itself is too complicated, the purpose of this paper is to analyze and evaluate its communication process. Therefore, we mainly use the CANoe platform to analyze the interaction data between the two parties. In this way, we can quickly achieve the experimental purpose of this paper and reduce the complexity of the research. We use the Communication Access Programming Language (CAPL) module that comes with the software to write the functions required by the protocol, including the message, byte length, and time of the sent message. CAPL mainly has the following functions:Simulate communication nodes, including time messages, periodic messages, and some repetitive messages;Simulate timed or network events for a node;The specific behavior of different time events can be simulated;Some network errors can be simulated to evaluate the associated error-proofing mechanisms;Provides test library functions for network testing.

Figure 15 shows the demand function code we wrote according to the CAPL function module. The main contents are: adding the corresponding protocol timer, Timer, and obtaining the related functions of the protocol type, message length, content, and content type.

At the same time, we combine the Security Manager function module of the simulation platform, which can add corresponding security functions according to the needs of users. According to the security scheme designed in Figure 9, we added the corresponding timestamp, random number and key, and other functions in the simulation process. Using the Trace function module of the CANoe platform, we can see the data transmission and transmission status of each time node in the process of transmitting messages in the protocol. As shown in Figure 16, the trace diagram of the protocol environment model designed for us, according to the real transmission rate of the vehicle Ethernet, the transmission speed of 100 Mbit/s is set. The main data period content includes the SOME/IP discovery service phase, message verification, and sending phase. The transmitted messages mainly include the Header and Payload parts of the protocol and data message packets.

### 7.2. Protocol Performance Analysis

This section will analyze the performance of our scheme based on the CANoe simulation results in Section 7.1 and the analysis tools of CPN Tools.

#### 7.2.1. Memory Usage Analysis of Design Scheme

According to the security scheme we designed, as shown in Figure 9, the message transmission process is mainly divided into the key negotiation process of identity authentication and the message transmission process. Table 7 shows the length occupied by each symbol used in the security scheme.

As shown in the table, we can see that the length of each identity ID occupies 1 byte, the length of the random number and timestamp occupies 4 bytes and 8 bytes, respectively, and the user identity key and session key both occupy 16 words. From this, we can conclude that the two communication requests and the two response requests occupy 30 and 48 bytes, as well as 78 and 63 bytes, respectively, and the total length of the bytes they occupy occupies 219 bytes. In addition, we use CANoe’s Detail View function module to further analyze the data in Figure 16, and we can find that the transmitted data DATA and service discovery data occupy 40 bytes and 102 bytes, respectively.

#### 7.2.2. Time Consumption Analysis of Security Solution

The security scheme designed in this paper is based on the symmetric encryption algorithm, which requires much less time than asymmetric encryption. The time consumed in the scheme is mainly divided into the time $T_{ge}$ consumed by generating data, such as identity ID and key; the time Ten and $T_{de}$ consumed by encryption and decryption; and the time consumed by verification $T_{ve}$. For the sender in the scheme, the time consumed includes one data generation time, one encryption and two decryption times, and one verification time, and the final time consumed is $T_{ge}$ + $T_{en}$ + 2$T_{de}$ + $T_{ve}$. For the receiver, the time consumed includes two data generation times, two encryption and two decryption times, and two verification times, and the final time consumed is 2$T_{ge}$ + 2$T_{en}$ + 2$T_{de}$ + 2$T_{ve}$. For the time consumed by the authentication center, there are one data generation time, two encryption and one decryption times, and one verification time. The final time consumed is 1$T_{ge}$ + 2$T_{en}$ + 1$T_{de}$ + 1$T_{ve}$, and the total time consumed by authentication is 4$T_{ge}$ + 5$T_{en}$ + 5$T_{de}$ + 4$T_{ve}$.

At the same time, we analyzed the data in Figure 16 and found that the load capacity consumed increases gradually over time, but the time consumption is mainly when the onboard equipment is started or stopped and also when the protocol is used to find remote services. It will consume a certain amount of time and load capacity. The communication process in our scheme is affected very little because the encryption and decryption of the communication process have little effect on the overall delay.

## 8. Summary and Outlook

This paper focuses on the security of SOME/IP, a service middleware protocol for in-vehicle Ethernet. Firstly, the message flow mechanism of the communication protocol interaction process is analyzed, and the protocol is modeled by CPN Tools. Then, the adversary model is introduced to attack the established SOME/IP model to verify the evaluation security. Afterwards, CPN internal tools are used to view the state space and analyze the state data [38]. It was found that there are three main attacks in the protocol: replay, spoofing, and tampering. For the attacks analyzed, this paper proposes a scheme that adds timestamps and random numbers to strengthen the authentication of the protocol, thereby providing reliable and secure transmission. The proposed reinforcement scheme is again modeled by CPN, and the same adversary model is introduced to verify whether our proposed reinforcement scheme is effective. By analyzing and comparing the state space results before and after the improvement, it is found that the new reinforcement scheme can effectively prevent the three existing attacks. Finally, we built a real vehicle Ethernet environment based on the simulation platform CANoe and conducted performance analysis and evaluation. However, in this paper’s research, we mainly consider the use of three attack methods to analyze the security of the protocol and do not carefully analyze other attack methods. In future research work, while enhancing the security of the protocol, we will try to add other attack methods to verify whether there are other types of security problems in the protocol.

## Figures and Tables

**Figure 1 sensors-22-06792-f001:**
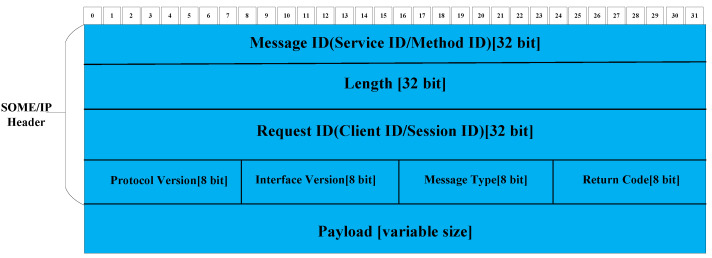
SOME/IP protocol data frame structure.

**Figure 2 sensors-22-06792-f002:**
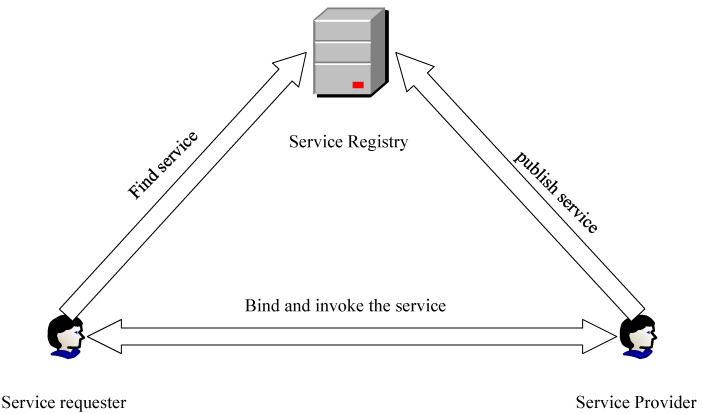
Middleware working principle.

**Figure 3 sensors-22-06792-f003:**
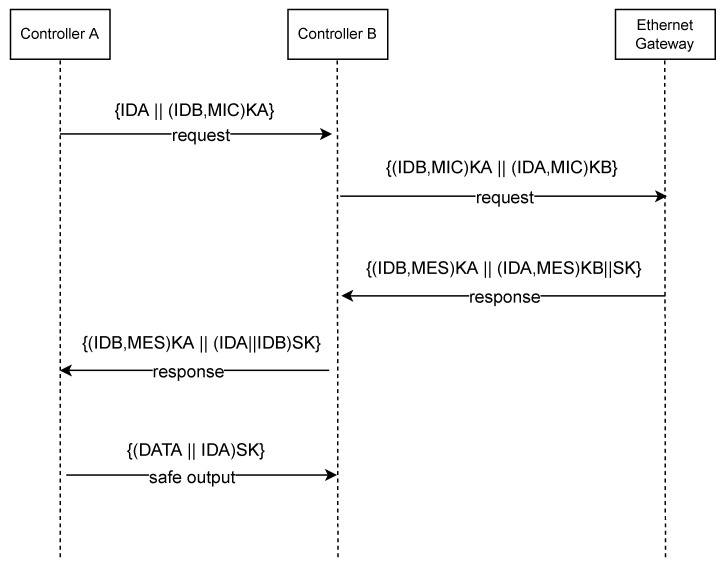
SOME/IP Message Flow.

**Figure 4 sensors-22-06792-f004:**
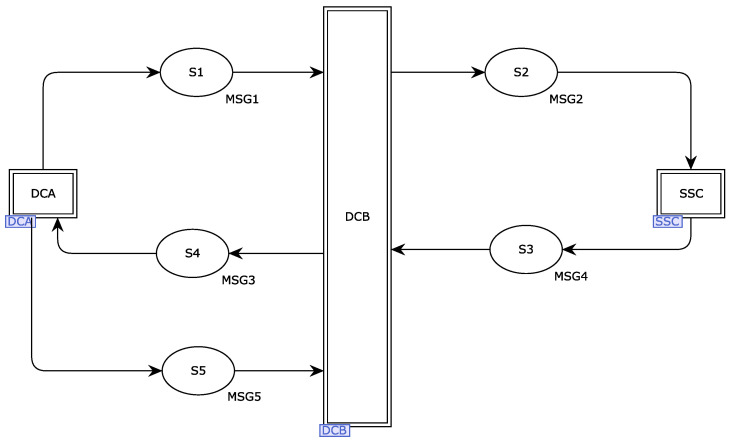
SOME/IP top-level model.

**Figure 5 sensors-22-06792-f005:**
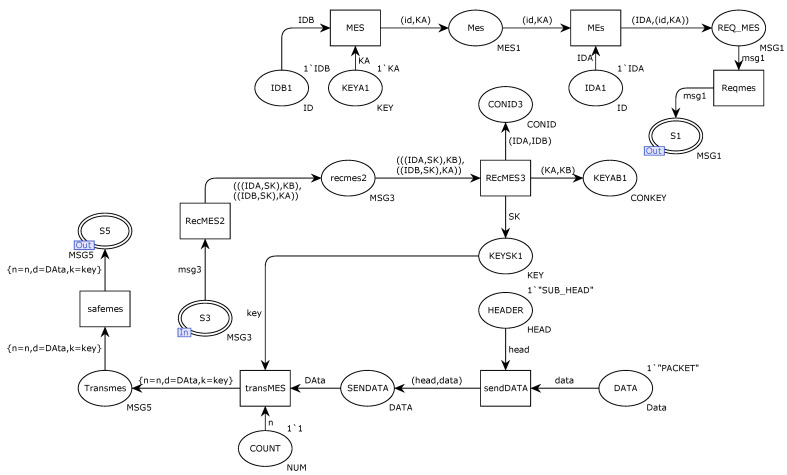
Internal model of alternative variation DCA.

**Figure 6 sensors-22-06792-f006:**
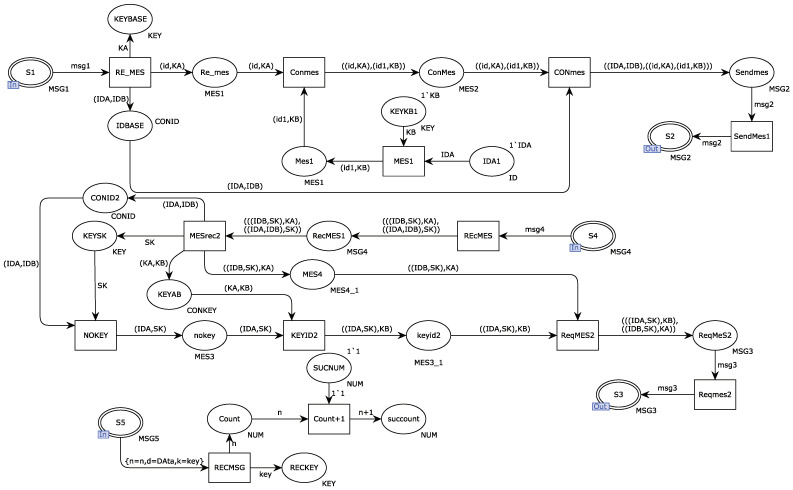
Internal model of alternative variant DCB.

**Figure 7 sensors-22-06792-f007:**
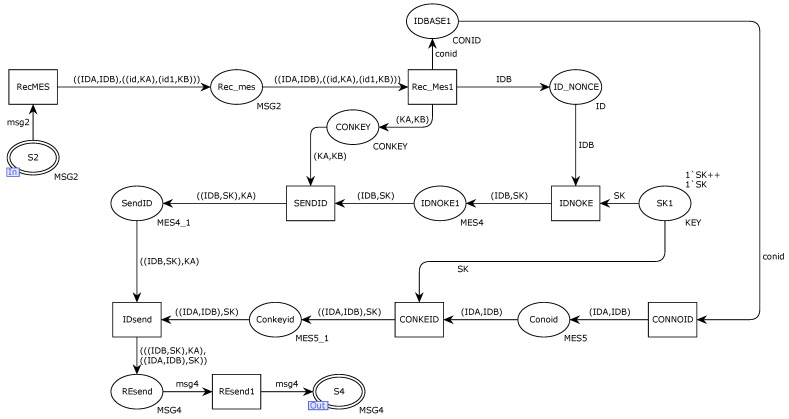
Internal model of alternative variation SSC.

**Figure 8 sensors-22-06792-f008:**
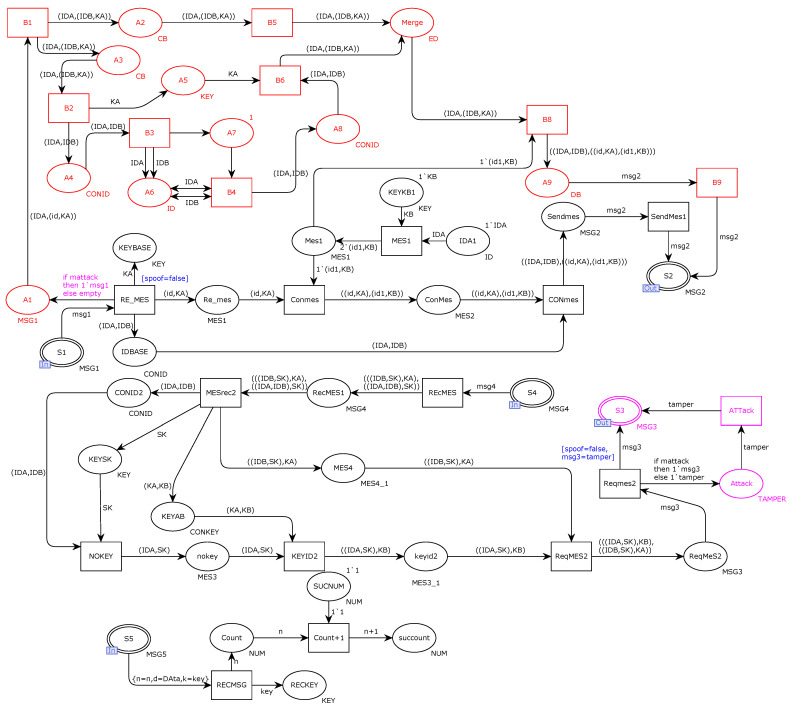
Attacker model.

**Figure 9 sensors-22-06792-f009:**
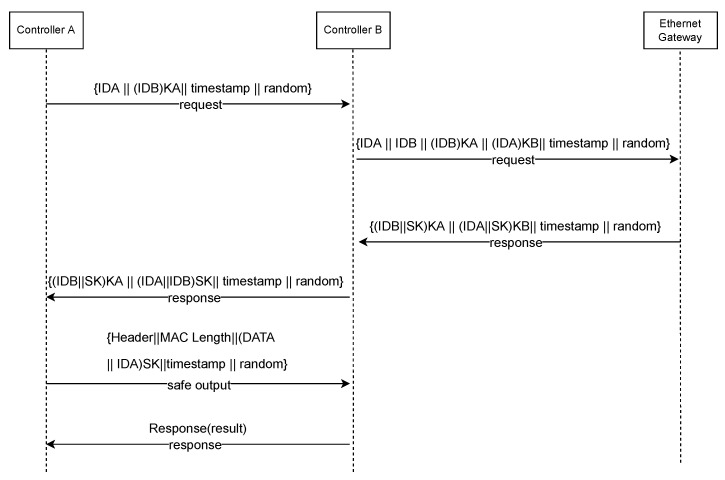
Message flow with improved protocol.

**Figure 10 sensors-22-06792-f010:**
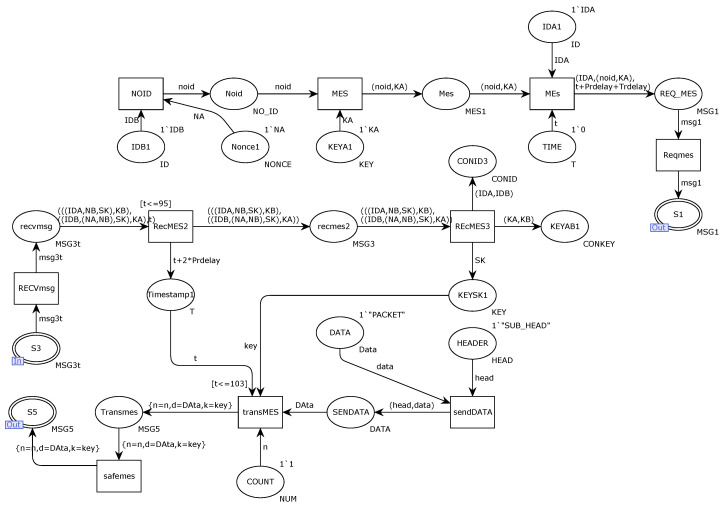
Internal model of the improved solution DCA.

**Figure 11 sensors-22-06792-f011:**
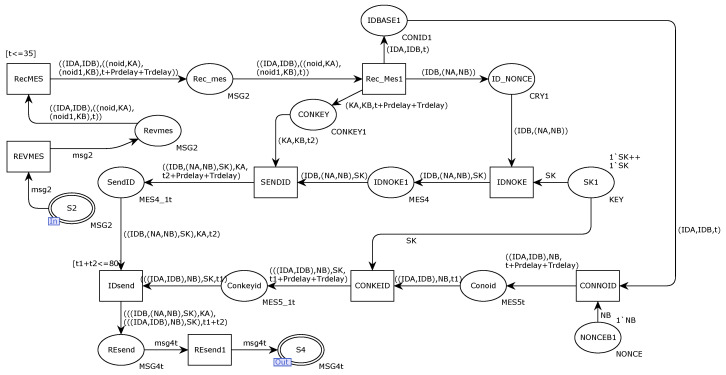
Internal model of improved solution SSC.

**Figure 12 sensors-22-06792-f012:**
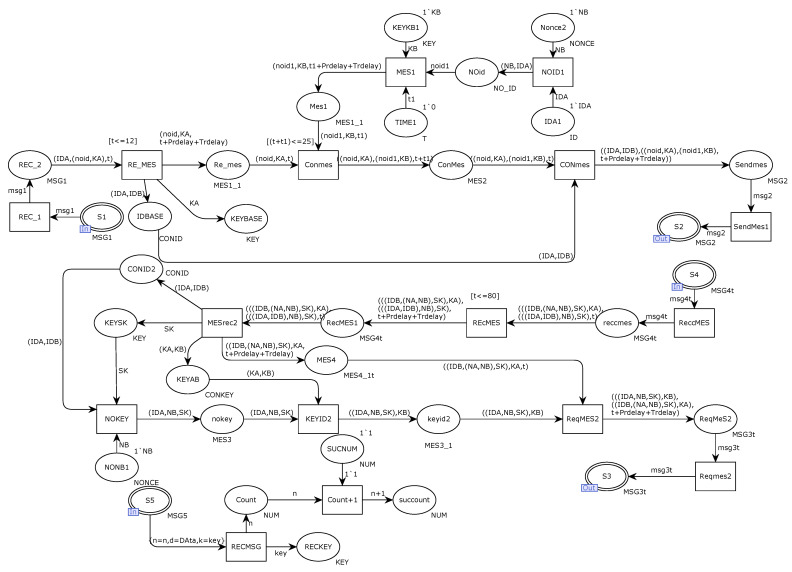
Internal model of the improved solution DCB.

**Figure 13 sensors-22-06792-f013:**
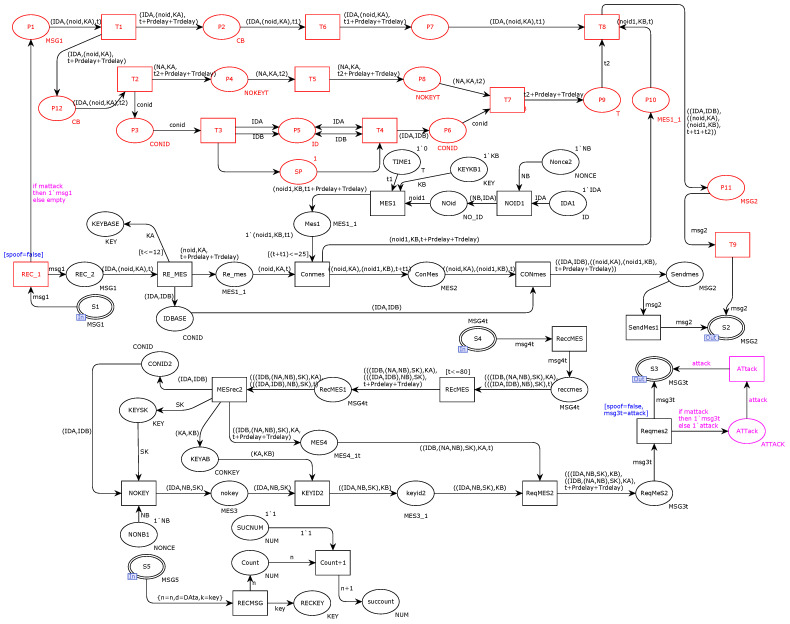
Improvement scheme to include the attacker model.

**Figure 14 sensors-22-06792-f014:**
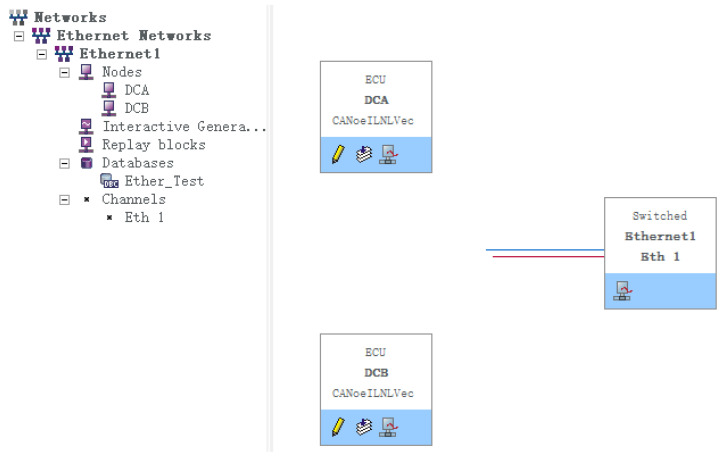
Protocol emulation environment.

**Figure 15 sensors-22-06792-f015:**
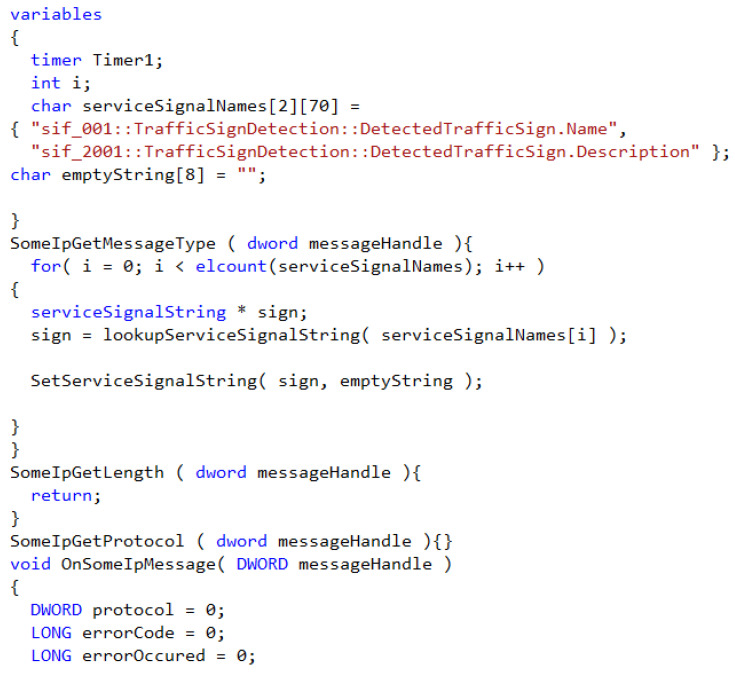
Function code.

**Figure 16 sensors-22-06792-f016:**
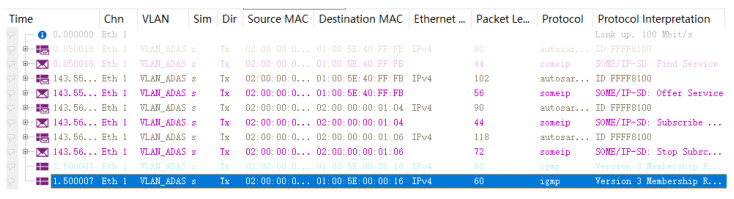
Trace communication data graph.

**Table 1 sensors-22-06792-t001:** Symbols and descriptions.

Symbols	Description
DCi	Device Controller
IDi	Device controller identity
SSC	Safety and security controllers
random	Random Number
Ki	The key generated by the device itself
SK	Temporary session key
request	Connection request
response	Connection Response
DATA	Required data after connection
timestamp	Timestamp
MIC	Integrity verification code
MES	Connect message
result	Return information

**Table 2 sensors-22-06792-t002:** State space analysis of the original model of SOME/IP protocol.

Type	Number
State space nodes	146
State space arc	333
SCC graph node	146
SCC graph arc	333
Live Transition Instances	0
Dead marking	1
Dead transition instances	0

**Table 3 sensors-22-06792-t003:** Comparison of model state spaces.

Type	Original Model	REY-ATK	TAR-ATK	SPF-ATK	Attack Model
State space nodes	146	16,354	256	702	22,702
State space arc	333	89,459	575	1531	104,371
SCC graph node	146	16,354	256	702	22,702
SCC graph arc	333	89,459	575	1531	104,371
Dead marking	1	8	2	6	11
Dead transition instances	0	0	0	0	0
Live Transition Instances	0	0	0	0	0

**Table 4 sensors-22-06792-t004:** State space comparison of three attack models.

Type	Before Improvement			New Scheme		
	REY-ATK	TAR-ATK	SPF-ATK	REY-ATK	TAR-ATK	SPF-ATK
State space nodes	16,354	256	702	11,100	272	236
State space arc	89,459	575	1531	39,052	536	460
SCC graph node	16,354	256	702	11,100	272	236
SCC graph arc	89,459	575	1531	39,052	536	460
Dead marking	8	2	6	2	1	1
Dead transition instances	0	0	0	0	0	0
Live Transition Instances	0	0	0	0	0	0

**Table 5 sensors-22-06792-t005:** Comparison of state spaces of security assessment models.

Type	Before Improvement	New Scheme
State space nodes	22,702	13,440
State space arc	104,371	48,920
SCC graph node	22,702	13,440
SCC graph arc	104,371	48,920
Dead marking	11	2

**Table 6 sensors-22-06792-t006:** Comparison of protocol analysis methods.

Scheme	Anomaly Detection	Attack Type	Intuitive Graphics	Verify Functional Correctness	State Space
Ref [31]	YES	NO	NO	YES	NO
Ref [33]	YES	NO	NO	NO	NO
Ref [21]	YES	NO	NO	YES	NO
Ref [34]	NO	YES	YES	NO	NO
Ref [35]	NO	YES	NO	YES	NO
Our Scheme	YES	YES	YES	YES	YES

**Table 7 sensors-22-06792-t007:** Symbols Occupied Length.

Symbols	Length (Bytes)
IDi	1
random	4
timestamp	8
Ki, SK	16
request1	30
request2	48
response1	78
response2	63
sum	219

## Data Availability

No data were used to support this study.

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
