# Peer review of "Security Analysis and Improvement of Vehicle Ethernet SOME/IP Protocol"

_sensors, 2022, doi:10.3390/s22186792_

Round 1
Reviewer 1 Report
In this paper, the authors investigate the security of the SOME/IP protocol by using formal analysis and propose an enhancement to it, to address some vulnerabilities they report. From a technical point of view the paper is of interest, but there are many improvements needed:
1. Firstly, the authors have to clarify were the CPN tool, which they use for modelling, can be retrieved from. On page 8 they point out to reference [17], but then on page 16 they point out to reference [25] as source for the tool. The authors have to clarify were this tool can be download and what classes of security protocols can it analyze.
2. Secondly, and more important, the authors have to be more specific about the attacks that they claim on this protocol. As presented in section 5.2 it is entirely unclear what attacks can the adversary mount on the protocol. They say “Table 3 shows that the introduced adversary model effectively attacks the request and response information transmission process of the original model”. Needless to say, this table presents no attack but the number of state space node. Attacks must be clearly presented and explained before proposing the enhancements.
3. Thirdly, there is huge amount of writing problems in this work, only some of the are summarized below:
- Incomplete sentences right from the abstract: "However, the connection between the intelligent" and "we introduce timestamps and random numbers to strengthen the”
- The acronym ICV, should stand for intelligent connected vehicles, not intelligent networked vehicle as the authors claim in the abstract
- Missing period: "and the Dolev-Yao adversary model", "After verification, the protocol is subject to three kinds of attack"
- Repeated “and”: "and the information and service applications and service platforms"
- Repeated words: "in in-vehicle networks", "for securing in-vehicle network security."
- Unexplained acronyms EFT, SOA
- Missing fields description of the data frame (Figure 1): Message ID, length, etc.
- Two repeated sentences: “The service requestor queries the registry when needed to obtain the relevant service interfaces and usage methods, completing remote invocation operations regardless of the development platform, programming language and software. The service requestor queries the registry when needed to obtain the relevant service interface and usage methods to complete remote invocation operations, regardless of the development platform, programming language and software architecture.”
- Table 1, which explains the terms used in the 8 steps of model needs to be introduced before these steps
4. Finally, I think that the related works section can be improved with more related works on in-vehicle security, as there are quite a number of works addressing security protocols for the CAN bus which are in-line with this work.
Author Response
Response to Reviewer 1 Comments
We gratefully thank the reviewers who took the time out of their busy schedules to provide constructive comments and helpful suggestions, which significantly improved the quality of the manuscript and allowed us to improve it. Each suggested revision and comment made by the reviewers was accurately incorporated and considered. The following is a point-by-point reply to the reviewers' comments, and the revisions are indicated.
Point 1: Firstly, the authors have to clarify were the CPN tool, which they use for modelling, can be retrieved from. On page 8 they point out to reference [17], but then on page 16 they point out to reference [25] as source for the tool. The authors have to clarify were this tool can be download and what classes of security protocols can it analyze.
Response 1: Thank you for your rigorous consideration. CPN Tools is an open tool that combines the advantages of editing, design, and simulation. Using its internal auxiliary tools, it is possible to simulate and debug the message transmission process of communication protocols in any field, and the markup language (ML) it uses is also easier to understand. On page 5 of our revised manuscript, we added to section 3.3: CPN Tools. This section introduces its features in detail, the reference [31] gives its official website, and the reference [32] introduces its background, features, and functions in detail.
(The above has been highlighted in red in the new manuscript.).
Point 2: Secondly, and more important, the authors have to be more specific about the attacks that they claim on this protocol. As presented in section 5.2 it is entirely unclear what attacks can the adversary mount on the protocol. They say “Table 3 shows that the introduced adversary model effectively attacks the request and response information transmission process of the original model”. Needless to say, this table presents no attack but the number of state space node. Attacks must be clearly presented and explained before proposing the enhancements.
Response 2: We gratefully appreciate your valuable suggestion. In our new manuscript, we added Section 5.1: Dolev-Yao Attacker Model. This section details the attack model we use, Dolev-Yao, which has the following attack characteristics:
- Illegal tampering: Attackers use illegal means to delete or modify legitimate data.
- Eavesdropping: The attacker listens to the messages transmitted by the user without being noticed.
- Illegal interception: The attacker illegally intercepts the transmission message and saves it without being discovered by the protocol.
- Replay: The attacker modifies the previously illegally intercepted message and sends it repeatedly to both parties.
In Section 5.2, we use the Dolev-Yao attacker model mentioned above for modeling, and Figure 8 shows the established attacker model.
In Section 5.3, Table 3 presents the state space data after adding attackers, and we use this data to analyze the three kinds of attacks in detail: eavesdropping, replay, and tampering with the protocol.
(The above has been highlighted in red in the new manuscript.).
Point 3: Thirdly, there is huge amount of writing problems in this work, only some of the are summarized below:
- Incomplete sentences right from the abstract: "However, the connection between the intelligent" and "we introduce timestamps and random numbers to strengthen the”
- The acronym ICV, should stand for intelligent connected vehicles, not intelligent networked vehicle as the authors claim in the abstract
- Missing period: "and the Dolev-Yao adversary model", "After verification, the protocol is subject to three kinds of attack"
- Repeated “and”: "and the information and service applications and service platforms"
- Repeated words: "in in-vehicle networks", "for securing in-vehicle network security."
- Unexplained acronyms EFT, SOA
- Missing fields description of the data frame (Figure 1): Message ID, length, etc.
- Two repeated sentences: “The service requestor queries the registry when needed to obtain the relevant service interfaces and usage methods, completing remote invocation operations regardless of the development platform, programming language and software. The service requestor queries the registry when needed to obtain the relevant service interface and usage methods to complete remote invocation operations, regardless of the development platform, programming language and software architecture.”
- Table 1, which explains the terms used in the 8 steps of model needs to be introduced before these steps
Response 3: We gratefully thank you for the precious time the reviewer spent making constructive remarks.
- We have carefully revised the writing and grammar issues that appeared in the abstract, and the revised Abstract reads as follows:
Abstract: The combination of in-vehicle networks and smart car devices has gradually developed into Intelligent Connected Vehicles (ICV). Through the vehicle security protocol, ICV can quickly realize the communication transmission. However, with the more frequent connections between smart in-vehicle devices and the network, the relationship between intelligent cars and external systems is becoming more and more complicated, and in-vehicle networks are gradually facing many security issues. Strengthening the security of in-vehicle protocols becomes particularly important. This paper uses the CPN model building method based on the colored Petri net theory to model the SOME/IP protocol of the vehicle Ethernet. The security protocol is formally verified and analyzed by combining it with the Dolev-Yao adversary model detection method. After verification, the protocol is subject to three attack vulnerabilities: replay, tampering, and deception. We introduce timestamps and random numbers to strengthen the protocol strengthen security. After the final analysis and verification, the improved scheme in this paper can effectively improve the security performance of the protocol.
- The acronyms that appear in the text have been revised, such as electrical fast transients(EFT)(Related work, Reference [9]), and service-oriented architecture(SOA)( Section 3.1).
- We have made careful changes to the repeated sentences in the proposal. The corrections are as follows: The service requester queries the registry when needed, obtains the relevant service interfaces and usage methods, and completes the remote invocation operation, which is not limited by the development platform, programming language, and software architecture. Maximize the independence between service consumers and service providers. (Section 3.2)
- We have modified the location of Table 1, which is now placed before the description steps, as required by the opinion. (Section 4.1)
(The above has been highlighted in red in the new manuscript.).
Point 4: Finally, I think that the related works section can be improved with more related works on in-vehicle security, as there are quite a number of works addressing security protocols for the CAN bus which are in-line with this work.
Response 4: We totally understand the reviewer’s concern. In response to your comments, we have added more analysis and literature on CAN bus security improvement in the Related Works section. For example, reference [7] [10] [29].
(The above has been highlighted in red in the new manuscript.).
Thank you again for your positive and constructive comments and suggestions on our manuscript.

Reviewer 2 Report
Based on the CPN modeling method of colored Petri net theory, this paper models the vehicle Ethernet SOME/IP protocol, and verifies the dolev Yao adversary model. The protocol is attacked by three kinds of attacks. In order to enhance security, this paper introduces time stamps and random numbers to enhance security. After the final analysis and verification, the improved scheme proposed in this paper can effectively improve the security performance.
1.There are repeated statements in the Abstract.
2.in the study of this paper, only considered the introduction of man-in-the-middle attacks to analyze the security of the protocol, and did not consider other attack methods.
3. The analysis of the state space data of the original model of the SOME/IP protocol and the state space data after adding an attacker can be more detailed.
4. The analysis and verification of the improvement scheme is less and can be more detailed.
Author Response
Response to Reviewer 2 Comments
We gratefully thank the reviewers who took the time out of their busy schedules to provide constructive comments and helpful suggestions, which significantly improved the quality of the manuscript and allowed us to improve it. Each suggested revision and comment made by the reviewers was accurately incorporated and considered. The following is a point-by-point reply to the reviewers' comments, and the revisions are indicated.
Point 1: There are repeated statements in the Abstract.
Response 1: We gratefully thank you for the precious time the reviewer spent making constructive remarks.
We have carefully revised the writing and grammar issues that appeared in the abstract, and the revised Abstract reads as follows:
Abstract: The combination of in-vehicle networks and smart car devices has gradually developed into Intelligent Connected Vehicles (ICV). Through the vehicle security protocol, ICV can quickly realize the communication transmission. However, with the more frequent connections between smart in-vehicle devices and the network, the relationship between intelligent cars and external systems is becoming more and more complicated, and in-vehicle networks are gradually facing many security issues. Strengthening the security of in-vehicle protocols becomes particularly important. This paper uses the CPN model building method based on the colored Petri net theory to model the SOME/IP protocol of the vehicle Ethernet. The security protocol is formally verified and analyzed by combining it with the Dolev-Yao adversary model detection method. After verification, the protocol is subject to three attack vulnerabilities: replay, tampering, and deception. We introduce timestamps and random numbers to strengthen the protocol strengthen security. After the final analysis and verification, the improved scheme in this paper can effectively improve the security performance of the protocol.
(The above has been highlighted in red in the new manuscript.).
Point 2: in the study of this paper, only considered the introduction of man-in-the-middle attacks to analyze the security of the protocol, and did not consider other attack methods.
Response 2: We gratefully appreciate your valuable suggestion. In this paper, the Dolev-Yao attacker model is used to analyze and verify the SOME/IP protocol. The scheme not only uses man-in-the-middle attacks but also involves replay attacks and eavesdropping attacks. In our new manuscript, we added Section 5.1: Dolev-Yao Attacker Model. This section details the attack model we use, Dolev-Yao, which has the following attack characteristics:
- Illegal tampering: Attackers use illegal means to delete or modify legitimate data.
- Eavesdropping: The attacker listens to the messages transmitted by the user without being noticed.
- Illegal interception: The attacker illegally intercepts the transmission message and saves it without being discovered by the protocol.
- Replay: The attacker modifies the previously illegally intercepted message and sends it repeatedly to both parties.
In Section 5.2, we use the Dolev-Yao attacker model mentioned above for modeling, and Figure 8 shows the established attacker model.
In Section 5.3, Table 3 presents the state space data after adding attackers, and we use this data to analyze the three kinds of attacks in detail: eavesdropping, replay, and tampering with the protocol.
(The above has been highlighted in red in the new manuscript.).
Point 3: The analysis of the state space data of the original model of the SOME/IP protocol and the state space data after adding an attacker can be more detailed.
Response 3: We gratefully thank you for the precious time the reviewer spent making constructive remarks. In response to your comments and suggestions, we have made the following improvements to our manuscript:
- For the state space of the original model, we analyze the consistency of the state space of the model in Section 4.3 of the article, and the purpose of verification is to verify whether the model we established has obtained the expected results through theoretical analysis. In Section 4.3.1, we analyze the possible expected results of the state-space data of the original model according to the message mechanism of the protocol. In Section 4.3.2, we analyze the state space data of the original model in more detail. The main contents are:
- Verify that there is no deadlock condition in the model.
- Determine the final state of the model.
- See if there are transitions that are always executing.
- Finds if there is a reachable path from one marker to another.
Finally, the data in Table 2 are analyzed in detail.
- For the state space analysis after adding the attacker model, we analyze the data change process in detail in Section 5.3 of the manuscript and summarize the existing security problems. In Section 5.3.1, we perform functional consistency security verification on the built model in order to verify whether the model consistency meets the requirements. Firstly, it is analyzed whether the state space nodes, state space directed arcs, and strongly connected nodes meet the test requirements under the three attacks, and then judge whether they meet the expected results. In Section 5.3.2, we analyze the state-space data obtained under the three attack modes by comparing the changes in the state data. We can preliminarily judge that the protocol model is affected by three attacks. In Section 5.3.3, through the above analysis. The protocol will be threatened by:
- If an attacker changes the value of the message serial number, the protocol will be authenticated incorrectly during authentication, which may lead to the authentication failure of the protocol.
- By intercepting the plaintext information, once the attacker obtains the message with the same sequence number, he can replay the message authentication code intercepted in the session to the receiver, which will affect the subsequent normal sending and receiving commands.
- Since there is a plaintext message when the two parties send messages, the attacker will eavesdrop on the communication data, analyze the illegally obtained data, and launch an attack on the node.
(The above has been highlighted in green in the new manuscript.).
Point 4: The analysis and verification of the improvement scheme is less and can be more detailed.
Response 4: We totally understand the reviewer’s concern. In view of the problem of less analysis of the improved scheme, in Section 6.4, we analyze in detail the obvious changes in the state space data of the three attacks under the improved scheme and the performance of our model under the mixed attack. Finally, the security realization of the improved scheme is analyzed. In Section 6.4.1, we compared the state space data of the three attack models before and after improvement in Table 4 and achieved our expected results. In Section 6.4.2, as shown in Table 5, three attack vectors are used simultaneously. We cannot cause effective attacks in our improved scheme, which proves that the reinforcement scheme can resist these three kinds of attacks and conform to the security mechanism of the protocol. In Section 6.4.3, the security performance of our improved scheme is analyzed according to the attacker model mentioned in Section 5.1 and the state space data in Table 4 and Table 5.
(The above has been highlighted in green in the new manuscript.).
Thank you again for your positive and constructive comments and suggestions on our manuscript.

Reviewer 3 Report
The authors need to describe more about how SOME/IP relates to IoV and the rationale for why it is used. At the same time, the authors need to describe in detail how to effectively identify man-in-the-middle attacks, and the defensive measures that can be taken.
Author Response
Response to Reviewer 3 Comments
We gratefully thank the reviewers who took the time out of their busy schedules to provide constructive comments and helpful suggestions, which significantly improved the quality of the manuscript and allowed us to improve it. Each suggested revision and comment made by the reviewers was accurately incorporated and considered. The following is a point-by-point reply to the reviewers' comments, and the revisions are indicated.
Point 1: The authors need to describe more about how SOME/IP relates to IoV and the rationale for why it is used. At the same time, the authors need to describe in detail how to effectively identify man-in-the-middle attacks, and the defensive measures that can be taken.
Response 1: Thank you for your rigorous consideration. In our new manuscript, we have made the following modifications and improvements to your comments and suggestions:
-- In our Introduction, we introduced more information about IoV, automotive Ethernet, and SOME/IP, detailing their relationship and the reasons for choosing SOME/IP. For example, The traditional Internet of Vehicles protocols: CAN, LIN, FlexRay, etc. have been proven. It is threatened by severe attacks such as network information tampering and virus intrusion. The emerging in-vehicle Ethernet is an organic combination of Ethernet technology and in-vehicle equipment, and its low complexity, high efficiency, and high-cost performance have attracted much attention in the field of car networking. SOME/IP, as a vehicle Ethernet application layer protocol, has also been studied and applied by many scholars. However, like traditional car networking protocols, in-vehicle Ethernet protocols face security issues. These network attacks will directly affect the regular use of in-vehicle equipment and cause user information leakage. In the face of more and more security problems of Internet of Vehicles devices, this paper will select the typical and commonly used vehicle Ethernet protocol SOME/IP as the analysis object. Strengthening the security inside the protocol has essential research significance in the industrial and academic fields. ( Introduction)
-- In Section 3.4, we detail several man-in-the-middle attacks, how to identify whether they are attacked, and the corresponding defense measures. In Section 3.4.1, we describe how to effectively identify man-in-the-middle attacks. For man-in-the-middle attacks, there are methods and tools available to identify and detect security threats. For example, when an attacker performs an attack, an additional time delay will be generated in the normal communication time. At this time, it is judged whether it is attacked by checking and calculating the difference in its sending response time. Secondly, we can analyze whether there are abnormal data packets. If we find that there is a big difference from normal data, we can suspect that we have been attacked. For many IoT devices, using intrusion detection is also a good choice. Capturing abnormal traffic data and analyzing the difference from normal data can be used to identify initial attacks. In Section 3.4.2, we provide some defenses against man-in-the-middle attacks. There are many ways to defend against an attack. For example:
- Improve inappropriate two-way authentication to prevent attacks. This method can prevent attackers from stealing internal information.
- Secondly, a more internal solid management system can be established. Once an attack is found, managers can quickly find and deal with it effectively.
- Enhancing the complexity of the verification required for authentication is also a good defense.
- Authentication by an authentication certificate issued by a professional authentication agency prevents eavesdropping and sniffing by attackers.
- A particular secure communication channel is established to verify the data exchanged between the two parties, and if a data leak occurs, it can be quickly detected.
- Finally, the user's safety awareness should be strengthened, and illegal operations should be paid attention to daily use.
(The above has been highlighted in blue in the new manuscript.).
Thank you again for your positive and constructive comments and suggestions on our manuscript.

Round 2
Reviewer 1 Report
The authors have addressed my comments, I have no further inquiries.
Author Response
Response to Reviewer 1 Comments
Thank you to the reviewers for your patience in reading our revisions, and I am glad that you are satisfied with the results of this improvement. Once again, we thank you for taking the time out of your busy schedule to provide us with all your constructive comments and helpful suggestions, which greatly improved the quality of the manuscript and allowed us to improve. Every suggested revision and comment made by the reviewers was accurately incorporated and considered every time. Finally, we wish you a smooth work and a happy life.
Reviewer 3 Report
No Comments.
Author Response
Response to Reviewer 3 Comments
Thank you to the reviewers for your patience in reading our revisions, and I am glad that you are satisfied with the results of this improvement. Once again, we thank you for taking the time out of your busy schedule to provide us with all your constructive comments and helpful suggestions, which greatly improved the quality of the manuscript and allowed us to improve. Every suggested revision and comment made by the reviewers was accurately incorporated and considered every time. Finally, we wish you a smooth work and a happy life.